# Genomic and Functional Variation of the Chlorophyll *d*-Producing Cyanobacterium *Acaryochloris marina*

**DOI:** 10.3390/microorganisms10030569

**Published:** 2022-03-06

**Authors:** Scott R. Miller, Heidi E. Abresch, Jacob J. Baroch, Caleb K. Fishman Miller, Arkadiy I. Garber, Andrew R. Oman, Nikea J. Ulrich

**Affiliations:** Division of Biological Sciences, University of Montana, Missoula, MT 59812, USA; heidi.abresch@umconnect.umt.edu (H.E.A.); jake.baroch@umconnect.umt.edu (J.J.B.); ckfmmt@gmail.com (C.K.F.M.); rkdgarber@gmail.com (A.I.G.); andy.oman@umconnect.umt.edu (A.R.O.); nikea.ulrich@umconnect.umt.edu (N.J.U.)

**Keywords:** *Acaryochloris*, chlorophyll, genomics, cyanobacteria, far-red photosynthesis, plasmid, horizontal gene transfer

## Abstract

The Chlorophyll *d*-producing cyanobacterium *Acaryochloris marina* is widely distributed in marine environments enriched in far-red light, but our understanding of its genomic and functional diversity is limited. Here, we take an integrative approach to investigate *A. marina* diversity for 37 strains, which includes twelve newly isolated strains from previously unsampled locations in Europe and the Pacific Northwest of North America. A genome-wide phylogeny revealed both that closely related *A. marina* have migrated within geographic regions and that distantly related *A. marina* lineages can co-occur. The distribution of traits mapped onto the phylogeny provided evidence of a dynamic evolutionary history of gene gain and loss during *A. marina* diversification. Ancestral genes that were differentially retained or lost by strains include plasmid-encoded sodium-transporting ATPase and bidirectional NiFe-hydrogenase genes that may be involved in salt tolerance and redox balance under fermentative conditions, respectively. The acquisition of genes by horizontal transfer has also played an important role in the evolution of new functions, such as nitrogen fixation. Together, our results resolve examples in which genome content and ecotypic variation for nutrient metabolism and environmental tolerance have diversified during the evolutionary history of this unusual photosynthetic bacterium.

## 1. Introduction

The marine cyanobacterium *Acaryochloris marina* is unique in its use of Chlorophyll *d* (Chl *d*) as its major photosynthetic pigment [1]. Although Chl *d* is nearly identical in structure to the ubiquitous Chl *a* of other cyanobacteria, algae, and plants, this pigment absorbs strongly at far-red wavelengths that are inaccessible to most oxygenic phototrophs. Because these wavelengths attenuate rapidly in water, *A. marina* appears to be restricted to shallow environments enriched in filtered far-red light, and its abundance has been observed to decrease with depth [2]. Nonetheless, Chl *d* and its derivatives can be abundant in marine coastal ecosystems [3], and *A. marina* is widely distributed in temperate and tropical saline environments, often in association with red algae or animals [1,4,5,6,7,8]. These cyanobacteria have also been confirmed, through either laboratory cultivation or environmental DNA sequencing, to occur in marine stromatolites [9], a saline lake epilithic biofilm [10,11], a microbial mat from a high-elevation brackish lake [12], and the aquatic plant rhizosphere of a heavily canopied freshwater stream [13].

Despite its broad distribution, the genomic diversity of *A. marina* remains poorly understood. This is because, until recently [8], genome data were available for only two laboratory strains [14,15]: the type strain MBIC110017 and strain CCMEE 5410, which was isolated from the Salton Sea, California. Similarly, we are largely ignorant regarding the extent of functional variation for traits that are expected to be important for *A. marina* life history, including nitrogen metabolism and salinity tolerance. Improving our understanding of this variation is essential for inferring the different ways that this bacterium may contribute to ecosystem processes in far-red enriched environments. We investigated this issue for our collection of 37 *A. marina* laboratory strains, including twelve newly isolated strains (Appendix A). The collection consists of epiphytes of red algae, strains associated with tunicates (MBIC11017, MU11, MU12, and MU13), and an epilithic strain (CCMEE 5410). We identified roles for both plasmid maintenance and loss, along with horizontal gene transfer (HGT), in the origins of metabolic diversity during *A. marina* evolution. We also report ecotypic variation for phenotypic traits, including salinity tolerance and nitrogen sources, particularly nitrogen fixation. Finally, we identified differences among strains with respect to the presence of an intact CRISPR-Cas system, which may be selectively maintained when foreign DNA-induced mortality is high but otherwise lost.

## 2. Materials and Methods

### 2.1. Laboratory Strain Isolation

Twelve new laboratory strains of *A. marina* were isolated as part of this study from the following sample collections (Appendix A): red algae on intertidal rocks at Hug Point State Park, Oregon (28 June 2017); red algae on a low tide sand flat at Wreck Beach, Vancouver, British Columbia (7 August 2018); red algae on intertidal rocks at Praia de Carcavelos, Portugal (24 October 2018); red algae from dock tires at Friday Harbor Laboratories, Washington (5 January 2019); and a red alga on an intertidal rock at Queroianella, Italy (30 December 2019). Strains were isolated from these environmental samples as previously described [8] and are available upon request. Briefly, algal samples were incubated at 20 °C in screw-cap vials containing 15 mL of IOBG-11 medium under low far-red light (~1.25 μmol photons m^−2^ s^−1^ from LED diodes with 710 nm peak emission). Clonal laboratory isolates were obtained from successful enrichments through repeated streaks and transfers on IOBG-11 agar plates.

### 2.2. Genomics

DNA was extracted using the DNeasy PowerBiofilm DNA extraction kit (Qiagen) following manufacturer instructions. Sample libraries were prepared and sequenced on an Illumina NextSeq 550 platform (150-bp paired-end) at the University of Pittsburgh Microbial Genome Sequencing Center. Sequence reads were trimmed of trailing low-quality bases and filtered based on read length and sequence quality with Trimmomatic version 0.36 [16], and draft genome assemblies were obtained with SPAdes version 3.12.0 [17] using manually optimized parameter settings to maximize the N50. Following the removal of contaminant contigs with Bandage version 0.8.1 [18] and Kraken 2 [19], assemblies were annotated by RAST [20]. Genome completeness was estimated with CheckM version 1.0.18 [21] and BUSCO version 4.1.2 [22]. Genome sequence data for strains described in this study are available at NCBI BioProject PRJNA649288.

### 2.3. Phylogenetics

A genome-wide phylogeny was reconstructed for the *A. marina* strains *A. thomasi* RCC1774 (NCBI accession NZ_PQWO00000000) and *Cyanothece* sp. PCC 7425 (NCBI accession GCA_000022045.1). A total of 1369 single copy groups of orthologous CDS were identified by OrthoFinder v2.2.7 [23] to create a concatenated alignment of protein sequences. We constructed a maximum likelihood tree with 1000 ultrafast bootstrap replicates [24] using IQ-TREE version 2.0 [25] according to the JTT+F+R5 model of sequence evolution selected by AIC in ModelFinder [26]. Additionally, a 16S rRNA gene phylogeny was reconstructed by maximum likelihood according to the TIM3+F+R2 model. For the *nif* dataset, maximum likelihood trees were reconstructed for *nifHDK* and 16S rRNA genes with a GTR+I+G model and 1000 bootstrap replicates, and a SH test [27] was used to test for topological congruence of trees.

### 2.4. Genomics and Bioinformatics

To identify *A. marina* core genes, we performed two local BLAST [28] approaches for each genome: (1) a blastn search of each CDS (protein coding sequence) in the *A. marina* reference strain MBIC11017 against each target genome assembly; and (2) a tblastn search of each protein sequence in a genome against the reference strain MBIC11017 genome. For (2), hits with <50% overlap between query and target and/or <50% sequence identity were removed. We used an E-value cut-off of 0.01 for both approaches. For each strain’s genome, BLAST outputs were merged, and a nonredundant gene set was then obtained with a custom Python script. Gene sets for all genomes were then combined, and a custom Python script was used to count how many genomes possessed a given gene.

KEGG analysis was performed with eggNOG [29]. Of CDS sequences in the inferred *A. marina* core genome, 2642 out of 2757 met the criteria of a minimum hit E-value of 0.001, a minimum sequence identity of 40%, and minimum query and subject coverages of 20%. FeGenie was used with default settings to identify iron metabolism genes [30], and a dendrogram was generated with Pvlcust [31], which hierarchically clustered normalized gene counts by Euclidian distance. Putative siderophore synthesis gene clusters were confirmed using AntiSMASH [32]. CRISPR arrays were obtained from RAST, and CrisprFinder [33] was used to extract individual spacers, which were queried by BLAST against plasmid and virus databases with CRISPRTarget [34].

### 2.5. Nitrogen Fixation and Salt Tolerance Experiments

Stock cultures were grown in 125 mL flasks containing 75 mL of ASN-III medium (https://www.atcc.org, accessed on 20 July 2020), which contains 0.43 M NaCl. Flasks were incubated at 25 °C under a 12:12 h light:dark cycle with a mean light intensity of ~25 μmol photons m^−2^ s^−1^ of cool white fluorescent light. For the salt tolerance experiment, each stock of growing cells was inoculated to an optical density at 750 nm (OD_750_) value of 0.01 in duplicate 125 mL flasks for each experimental salinity treatment (75 mL of ASN-III medium modified to contain final NaCl concentrations of 0, 0.2, 0.43, 0.5, 0.62, 0.8, or 1.6 M). A similar approach was taken for the nitrogen fixation experiment. For the +N control treatment, stock cells were inoculated into fresh ASN-III; for the -N treatment, cells were inoculated into ASN-III-N medium (ASN-III without sodium nitrate added) after rinsing with ASN-III -N. For the -N rinse, cells were first pelleted in a 2 mL microfuge tube and the supernatant removed; then, 1 mL of ASN III -N media was added to the tube, the tube was vortexed, and the process repeated until cells had been rinsed three times. Experimental flasks were incubated at randomized positions in a growth chamber under the same environmental conditions as above. Every 48 h, the OD_750_ of a 2 mL homogenized culture sub-sample was measured for each flask with a Beckman Coulter DU 530 spectrophotometer (Indianapolis, IN, USA), and flasks were randomly re-positioned in the chamber. Growth was measured as the increase OD_750_ over time. Exponential growth rates were estimated for cultures that sustained growth for at least three population doublings. All statistical models were analyzed with JMP version 14.2 (SAS Institute Inc., Cary, NC, USA).

## 3. Results and Discussion

### 3.1. A. marina Phylogeny and Core Genome

We took a phylogenomics approach to develop a better understanding of *A. marina* genome variation and functional diversity. Our data set consisted of genome data for 37 *A. marina* laboratory strains from diverse geographic locations (Figure 1; Appendix A). Together with previously available data [8,14,15], this included genome assemblies for twelve newly isolated strains. Among these are the first *A. marina* strains from the Salish Sea (San Juan Island, WA, USA and Vancouver, BC, Canada), the eastern Atlantic Ocean (Carcavelos, Portugal), and the Mediterranean Sea (Queroianella, Italy). Most are draft assemblies constructed from Illumina sequence data but are virtually complete (Appendix A).

Our phylogeny provides fresh insights on *A. marina* diversification (Figure 1). As we previously reported [8], *A. thomasi* strain RCC1774, which produces Chl *b* instead of Chl *d*, was sister to the *A. marina* clade. *A. marina* strains isolated from the west coast of North America (sites FH, HP, S, MSP) were generally very closely related but did not cluster exclusively by geographic location (Figure 1), which implies a history of migration along the coast. New strains from Portugal and Italy provided a first look at European *A. marina* diversity, which revealed close connections with strains from both the western and eastern Pacific. Notably, Portugal strains P4 and P9 belonged to the Pacific West Coast clade (and were identical in 16S rRNA gene sequence to these strains). By contrast, the Italy strain I2.1 was sister to strains isolated from Awaji Island, Japan. Finally, although more data will be required to better resolve *A. marina* global biogeography, it is already clear that distantly related lineages can co-occur (for example, MU03 and MU04 from Hokkaido, Japan; Figure 1).

Tunicate-associated strains (Figure 1) formed part of a clade isolated from Palau (MBIC11017) and Okinawa (MU11, MU12, MU13). All were derived from the surface of didemnid tunicates that have a symbiotic association with the cyanobacterium *Prochloron*, although members of this clade can also be algal epiphytes (e.g., MU10). *A. marina* DNA has also been detected from a sponge sample collected in Palau [35]. Whether animal-associated *A. marina* are restricted to subtropical/tropical environments is not currently known.

Our data set also informs our understanding of the *A. marina* core genome. We considered a gene to belong to the core if it was detected in at least 36 of the 37 genomes. 2757 protein-coding genes belonged to the core (Figure 2; Appendix A; *N* = 2258 genes for a strict core of all 37 genomes). Few plasmid genes (*N* = 6) in the *A. marina* MBIC11017 reference genome are in this core, although they make up ~22% of its genome [14]. The majority of these were annotated as hypothetical proteins, with the exception of an RNA-binding protein and a phage integrase. This observation corroborates the earlier proposal that gene content of *A. marina* plasmids is highly dynamic and diverges rapidly between genomes [15]. 1502 core genes were assigned a KEGG Orthology ID, with 48 complete KEGG modules (Appendix A; 35 additional modules are missing only one block).

529 genes in the *A. marina* core were not present in the outgroup strains (Figure 2; Appendix A). This set of “unique core” genes potentially includes the unidentified gene(s) responsible for Chl *d* synthesis. Although Chl *d* is produced from Chl *a* and requires molecular oxygen [36], the identity of this enzyme(s) remains a longstanding puzzle in *Acaryochloris* biology. Yoneda et al. [37] identified 23 candidates from three oxygenase families, many of which were not co-expressed with other chlorophyll biosynthesis genes. These include the Rieske oxidase (RO; [38]), antibiotic biosynthesis monooxygenase (ABM) [39], and cytochrome P450 [40] families. Our core analysis excludes several of these, which do not belong to the *A. marina* core: AMB genes AM1_3889 and AM1_4175 (strain MBIC11017 gene designations), RO genes AM1_0159 and AM1_A0067, and P450 genes AM1_3563, AM1_4161, and AM1_5780. In fact, only one of the 23 genes belongs to the *A. marina* unique core: RO gene AM1_2905, annotated as a pheophorbide *a* oxygenase (PAO). PAO is involved in Chl *a* degradation through the oxygenolytic opening of the porphyrin ring [41]; the PAO enzyme in *A. marina* MBIC11017 is likely AM1_0031, which is the ortholog of PAO in *A. thomasi* RCC1774 (61% amino acid identity). AM1_2905 appears to have been acquired by an ancient HGT event prior to the diversification of extant *A. marina*. Whether this PAO homolog has evolved a new function in chlorophyll metabolism remains to be tested.

The estimated total number of genes (and genome size; Appendix A) varies greatly among *A. marina* genomes (~5800–9600) but in all cases greatly exceeds the inferred core. Below, we focus on several examples of variation in gene content among members of the *A. marina* radiation.

### 3.2. Differential Retention of Ancestral Plasmid Gene Content Contributes to A. marina Functional Variation

Several notable genes that are observed in *A. thomasi* RCC1774 and were likely present in the *A. marina* common ancestor appear to have been differentially retained or lost during *A. marina* diversification (Figure 1). For example, many *A. marina* strains possess a second set of ATP synthase (ATPase) genes first reported for strain MBIC11017 [14]. These annotated sodium-transporting ATPase (Na^+^-ATPase) genes are located on a conserved ~100 kbp block of plasmid sequence in the closed (or nearly closed) genomes of strains MBIC11017 (plasmid pREB4), CCMEE 5410, and S15 (Figure 3A,B). The genes are homologous to and share conserved gene order with the Na^+^-ATPase operon of the halotolerant cyanobacterium *Aphanothece halophytica* [42]. The *A. halophytica* Na^+^-ATPase is a sodium pump with increased activity at increasing NaCl concentrations [43], and heterologous expression conferred enhanced salt tolerance in the freshwater cyanobacterium *Synechococcus* PCC 7942 [42]. ATPase selectivity for sodium ions versus protons is controlled by the protein environment in the vicinity of the ion-binding site of the ATPase c-ring (AtpH) [44,45]. In *A. marina*, the amino acids of this binding site are conserved with *A. halophytica* [42], which suggests that these genes indeed encode a Na^+^-ATPase that may be involved in salt tolerance (see below).

The putative Na^+^-ATPase completely co-varies with genes encoding a bidirectional NiFe-hydrogenase (*hoxEFUYH*) and its associated maturation proteins that are found on the same plasmid as the *atp* genes (Figure 1 and Figure 3A,C). Many cyanobacteria possess a bidirectional hydrogenase, but it is most commonly observed in strains isolated from environments that periodically experience microoxic or anoxic conditions, including microbial mats, salt marshes, and intertidal habitats [46]. The physiological role of this enzyme in cyanobacteria is still debated. Activity is biased toward hydrogen production in *Synechocystis* PCC 6803 [47], and proposed functions include as an electron sink in fermentation [46] and/or photosynthesis [48], or as a general regulator of the cellular oxidation state [49]. In *Synechocystis* PCC 6803, electrons for hydrogen production are provided by reduced ferredoxin from photosystem I and pyruvate ferredoxin oxidoreductase (PFOR) activity, respectively, and a functional hydrogenase is essential for growth under anaerobic, nitrate-limiting conditions [50].

Conserved plasmid content also includes a glycogen phosphorylase (MBIC11017 nucleotide positions 164,576–167,203 in Figure 3) and PFOR genes (MBIC11017 positions 209,710–213,333). We propose that the plasmid may therefore provide a module for maintaining redox balance under fermentative conditions similar to the model of Khanna and Lindblad [51], with (1) glycogen catabolism to pyruvate, followed by (2) the oxidation of pyruvate to acetyl-CoA and reduction of ferredoxin by PFOR, and (3) hydrogen production via electron donation from ferredoxin to hydrogenase. This is in accord with the observation that *hox* expression in *A. marina* MBIC11017 is induced under anoxic, low light, and far-red light conditions [52,53]. Expression may potentially be controlled by conserved sensory and regulatory genes on the plasmid, which include a RpoD family sigma factor and a PAS domain S-box protein with histidine kinase and response regulator domains. PAS family proteins act as oxygen, redox status, or light sensors [54], and the gene includes a GAF domain with homology to FhlA (*E* = 4 × 10^−11^). FhlA plays a role in the transcription of the formate hydrogenlyase complex, which is responsible for fermentative hydrogen production in *E. coli* [55]. Whether there is a metabolic connection between the hydrogenase and the Na^+^-ATPase remains to be investigated. In addition to a possible halotolerance function, nitrogen-starved cells of *Synechocystis* PCC 6803 use ATPase to produce a sodium motive force that maintains sufficient levels of ATP for survival during metabolic dormancy [56]. Both sodium bioenergetics and glycogen catabolism are important during the resumption of growth in *Synechocystis* PCC 6803 following nitrogen starvation [56], and future work could address if hydrogenase activity also contributes.

The distribution of *atp* and *hox* genes (Figure 1) indicates a dynamic history of retention and loss of the entire plasmid that bears these genes over millions of years of *A. marina* diversification [11]. The persistence of plasmids is a “paradox” [57] that has long puzzled biologists. Since they often impose a fitness cost on the host (e.g., via an expression burden), plasmids are generally expected to be lost over evolutionary time through either segregational instability or by the integration of beneficial plasmid genes into the chromosome [58]. Yet, it is possible for plasmids to be stably maintained not only in selectively favored environments but also in the absence of selection [59,60]. This can occur through compensatory evolution that reduces the fitness cost to the host or by plasmid mechanisms that promote persistence, such as a partitioning system to prevent segregational loss, conjugative transfer, or an addiction module such as a toxin-antitoxin system [61]. While the conserved core of the *atp*/*hox* plasmid lacks conjugation machinery and addiction modules, it does include plasmid partition genes *parAB*.

The role of selection in *atp*/*hox* plasmid persistence remains to be determined, as does whether plasmid loss is neutral or favored in particular environments. However, it is clear that these ecological and evolutionary dynamics can operate at a local scale. For example, strains with or without the plasmid co-occur at Hug Point, Oregon (Figure 1), where samples were collected at low tide from exposed rock and tide pools. One avenue for future research would be to investigate whether cells that have retained the plasmid are more likely to experience periods of anoxia, salt stress, and/or metabolic dormancy, perhaps due to microenvironmental differences in the probability or duration of tidal exposure.

### 3.3. HGT and the Evolution of Nitrogen Metabolism

*A. marina* strains have the coding potential to assimilate a variety of nitrogen (N) sources, including ammonium, nitrate, nitrite, urea, and cyanate. In addition, some *A. marina* genomes contain idiosyncratic N metabolism genes beyond these core capacities. For example, in addition to an assimilatory ferredoxin nitrite-reductase, one *A. marina* clade possesses transposase-flanked *nirBD* genes for nitrite reductase (Figure 1) that were likely acquired by HGT from another cyanobacterium based on sequence similarity. In enterobacteria, NirBD is a dissimilatory enzyme that detoxifies nitrite produced by respiratory nitrate reduction under anoxic conditions [62]. Its role in *A. marina* is not clear; however, in strain CCMEE 5410, expression of these genes is similar to assimilatory ferredoxin nitrite-reductase in N deficient versus N replete conditions [37], which suggests that it may be an alternative assimilatory nitrite reductase.

N fixation of atmospheric N_2_ to ammonium is an important source of biologically available N. The ability to fix N has been previously reported for one strain of *A. marina* (HICR111A) [63]. Like many unicellular, N-fixing cyanobacteria, strain HICR111A fixes N during the dark period [63], when nitrogenase-inhibiting oxygen is not produced by photosynthesis. N fixation genes are absent from the genomes of strains MBIC11017 and CCMEE 5410, and N fixation ability in HICR111A has been proposed to have been acquired by HGT [63]. However, the evolutionary history of this trait, including whether the common ancestor of *A. marina* possessed *nif* genes, has not been fully resolved due to limited taxon sampling.

Our new phylogeny provides fresh insights into this history. We have identified a clade of epiphytic *A. marina* near the base of the phylogeny that also have N fixation genes (Figure 1). Members of this clade can grow with N_2_ as the sole N source at the same rate as on nitrate (Figure 4A). Although genome data are not publicly available for HICR111A, we believe that this strain likely belongs to this clade as well, based on a 16S rRNA gene tree (Appendix A), the *nif* gene sequence similarities of these strains, and the conservation of genes flanking the *nif* region among genomes [63].

There are two alternative hypotheses for how these strains obtained *nif* genes. The first is that they were vertically inherited from the *A. marina* common ancestor, with subsequent gene loss in other descendant strains. If this were the case, then we predict that the *nif* phylogeny will resemble the species tree inferred with 16S rRNA gene data. The second is that the most recent common ancestor of this clade acquired the *nif* genes more recently by HGT, in which case the *nif* and species trees will differ in topology. To test these hypotheses, we inferred maximum likelihood trees for both *nifHDK* (encoding the nitrogenase complex proteins) and 16S rRNA genes. In the species tree, the sister group of *A. marina* included *Cyanothece* PCC 7425 (Figure 4B), as has been observed in genome-wide phylogenies (Figure 1) [8,64]. By contrast, the most closely related sequences of *A. marina nifHDK* genes were from *Leptolyngbya* PCC 7375, with *Cyanothece* PCC 7425 genes more distantly related (Figure 4B). The topologies of the two trees were significantly different from each other, as determined by an SH test (−Δln L = 310.4, *p* < 0.0001) [27]. This indicates that the *A. marina* common ancestor did not possess *nif* genes; rather, these genes were obtained later by the ancestor of the GR1/MU08/MU09 clade (Figure 1) via HGT from another cyanobacterium.

Finally, N storage pools vary among strains. The genes required to produce and degrade cyanophycin, the primary N storage compound of many cyanobacteria [65], is absent in all *A. marina* strains except strain WB4 (Figure 1). The amino acid sequences of cyanophycin synthetase and cyanophycinase in WB4 have high similarity (83% and 79%) with *A. thomasi* RCC1774, indicating that these genes were likely present in the *A. marina* common ancestor but lost by other strains. Phycobilisomes are the primary light-harvesting complexes of most cyanobacteria and can account for approximately 50% of cell soluble protein [66,67]; however, they are quickly degraded during N limitation to provide N for other functions [68]. We have previously reported that the *A. marina* common ancestor had lost the genes for phycobiliprotein production and degradation but were uniquely reacquired by strain MBIC11017 [8]. Even with the addition here of twelve new *A. marina* genomes, MBIC11017 is still the only known strain to produce phycobiliproteins (Figure 1).

### 3.4. Ecotypic Variation in Salt Tolerance

All of the *A. marina* strains in our collection were isolated from different zones along tidal gradients, from the upper intertidal to brackish or saline subtidal environments (Appendix A). Consequently, they are expected to have historically experienced different frequencies and durations of stress associated with tidal activity, including the degree of exposure to both hyper- and hyposaline conditions. The salt tolerance range of marine algae (on which *A. marina* often grows as epiphytes) is generally broader for upper intertidal taxa than for subtidal species [69,70]. We therefore predicted that *A. marina* strains isolated from the upper intertidal zone would exhibit broader salinity tolerance than other strains.

To test this, we assayed growth at a range of NaCl concentrations for six *A. marina* strains. Strains S1 and S15 were isolated from upper intertidal and subtidal habitat, respectively, along a tidal gradient on the northern California coast (Shelter Cove; Appendix A).

Like S1, HP9 was isolated from the upper intertidal zone (Hug Point, OR). Strain FH11, which is very closely related to the above strains (Figure 1), was isolated from a submerged estuarine environment (San Juan Island, WA) that has a salinity that is approximately 85% of that of seawater (30 ppt; https://www.bco-dmo.org/dataset/775732, accessed on 26 May 2021). Strains MBIC11017 and CCMEE 5410 are from submerged habitats in the tropical western Pacific [1] and the Salton Sea, CA (34 ppt salinity at time of collection; [11]), respectively.

As predicted, upper intertidal strains S1 and HP9 exhibited the broadest range of NaCl concentration that could support sustained growth (0.2–0.8 M; Figure 5; all six strains bleached at 0 M and 1.6 M NaCl). S1 and subtidal strain S15 did not differ significantly in growth rate at 0.2 M and 0.43 M NaCl; however, S1 grew faster than S15 at higher NaCl concentrations (FDR-adjusted *p* = 0.08 at 0.5 M, *p* < 0.01 at 0.62 M, and only S1 could grow at 0.8 M). The lower salt tolerance of S15 was expected given its subtidal origin. Estuarine strain FH11 was even less halotolerant, with optimal growth observed at lower NaCl concentrations (*p* = 0.38 for the comparison between 0.2 M and 0.43 M) and particularly low growth rates at NaCl concentrations greater than 0.43 M (including no sustained growth at 0.8 M). Other strains from less variable salinity environments grew well at intermediate NaCl concentrations but did not grow at either the lowest salinity (MBIC11017) or highest salinity (CCMEE 5410) treatments. We conclude that *A. marina* halotolerance generally matches local environmental conditions.

In cyanobacteria, halotolerance involves both the export of sodium ions and the production of compatible solutes to maintain osmotic balance under conditions of high salinity; however, other key aspects of tolerance, including how cells recognize salt stress signals and regulate salt acclimation, is not well understood [71]. Understanding the observed differences in halotolerance among *A. marina* strains is therefore not possible from a consideration of gene content alone. Nonetheless, it is instructive to briefly discuss how differences in gene content may contribute to these phenotypic differences.

*A. marina* strains vary in terms of the types and number of systems for removing sodium ions. For example, strain S1 has three CPA1 family Na^+^/H^+^ antiporter genes, compared with one each in the close relatives S15, FH11, and HP9. Most strains also have one copy of the *mrpCDEFGAB* gene cluster, a multi-subunit CPA3 family Na^+^/H^+^ antiporter. However, the *mrp* cluster is absent in the FH1 genome, whereas strains S15, S7, HP8, and HP11 have two copies. Recent duplicates in *A. marina* are often found on different genetic elements [15], and we confirmed that duplicated Mrp genes are on a plasmid (contig 10, nucleotide positions 263,524–267,402) for strain S15. The homologous *A. halophytica* cluster complements a salt-sensitive mutant of *E. coli* [72]; in *Anabaena* PCC 7120, *mrpA* is upregulated in response to increasing NaCl, and its inactivation by transposon mutagenesis results in a salt-sensitive phenotype [73]. Finally, as discussed above, many strains have plasmid-encoded genes for a sodium-transporting ATPase (Figure 1).

Strains also exhibit variation in genes involved in the production or import of compatible solutes. All possess *ggpP* and *ggpS* genes required to synthesize glucosylglycerol, a common compatible solute of marine cyanobacteria [71]. In addition, many strains also encode an ABC transporter in the osmoprotectant uptake (Opu) family of compatible solute transporters [74] with unknown substrate specificity (Figure 1).

### 3.5. Iron Metabolism

Iron is an important nutrient for most organisms but is often limiting in oxic environments, where it is insoluble [75]. Consequently, bacteria have evolved mechanisms for scavenging and storing iron when it is rare. This includes the ability to synthesize and/or import siderophores, low molecular weight compounds that chelate iron with high affinity [76]. The ability to acquire iron may vary among organisms depending on iron availability in the environment. We used FeGenie [30] to identify genes related to iron metabolism in *A. marina* genomes. Strains clustered into two groups based on iron gene content that do not closely reflect the *A. marina* phylogeny (Figure 6 and Appendix A). Strains with more genes related to iron metabolism tend to be enriched in genes involved in the regulation of iron assimilation, siderophore transport, and, often, siderophore synthesis (Figure 6; Appendix A).

All strains possess annotated siderophore import genes (though they vary in number). By contrast, genes involved in the synthesis of siderophores are scattered throughout the phylogeny and appear to have been independently acquired at least three times during *A. marina* diversification (Figure 1). In addition to the MBIC11017 cluster, the most common siderophore gene cluster was observed in strains isolated from the U.S. West Coast (FH, HP, and MSP sites) and in the Japanese strain NIES2412. The closest matches in the antiSMASH database of microbial secondary metabolites [32] were paenibactin and myxochelin. Homologs of these genes were detected by BLAST in an assembly of the cyanobacterium *Leptolyngbya* from metagenome data obtained for a South African peritidal stromatolite (NCBI BioProject PRJNA612530, assembly SM1_1_3) [77]. It is not clear whether this cluster was inherited from a common ancestor or independently acquired. A third cluster in strains MU06, MU07, and MU03 exhibited the closest similarity to cupriachelin in the antiSMASH database and exhibit homology to genes in *A. thomasi* RCC1774 (48–68% amino acid identity between the latter and MU06).

Enhanced iron gene content may be selectively favored in low iron environments. For example, the genome of strain MBIC11017, which was isolated from an iron-poor region of the Pacific (1), contains both plasmid-encoded iron transport gene duplicates and a transposase-flanked cluster of genes [78] that is homologous to characterized siderophore synthesis genes in the cyanobacterium *Anabaena* PCC 7120 [79]. These genes are associated with higher rates of iron acquisition and growth under conditions of low iron availability compared with strain CCMEE 5410 [78], which is a member of the “low iron” gene content cluster (Figure 6).

### 3.6. Complex History of CRISPR-Cas Systems

*A. marina* genomes also vary with respect to the presence of CRISPR-Cas systems (Figure 1), all of which are either type I or type III class 1 systems (Table 1; Figure 7). The number of CRISPR arrays varied among these genomes (1–8 arrays), as did the total number of spacers (17–140 spacers; Appendix A). The majority of spacers were derived from uncharacterized viral sequences, although plasmid sequences were also detected (Appendix A).

The scattered distribution of CRISPR-Cas among extant *A. marina* strains (Figure 1) suggests a complicated history of gains, retentions, and losses. Two of the systems (the I-E and one of the III-B systems) are observed in strains from geographically distant locations (both cases involve western North America and Japan) among distantly related strains (Figure 1). Both exhibit a highest sequence identity with distantly related cyanobacteria (*Romeria gracilis* (I-E) and *Aphanocapsa montana* (III-B), respectively) and therefore clearly have been acquired by HGT.

There is both theoretical and empirical support for the idea that organisms with functional CRISPR-Cas systems obtain protection from phages and plasmids at the cost of a reduction in the ability to acquire potentially beneficial genes by HGT [80,81]. We speculate that such a trade-off may have played out during *A. marina* diversification. An intact CRISPR-Cas may be selectively favored under conditions of high foreign DNA-induced mortality but otherwise lost; its absence may then result in a greater influx of new genes by HGT.

## 4. Conclusions

Laboratory strain collections and genomic resources are essential for developing an understanding of the genetic and phenotypic variation of recently discovered microorganisms. Our study illustrates several ways by which genome content and ecotypic variation for nutrient metabolism and environmental tolerance have evolved during *A. marina* diversification. It further identifies phenotypic variation of interest for future comparative physiology and omics investigations.

## Figures and Tables

**Figure 1 microorganisms-10-00569-f001:**
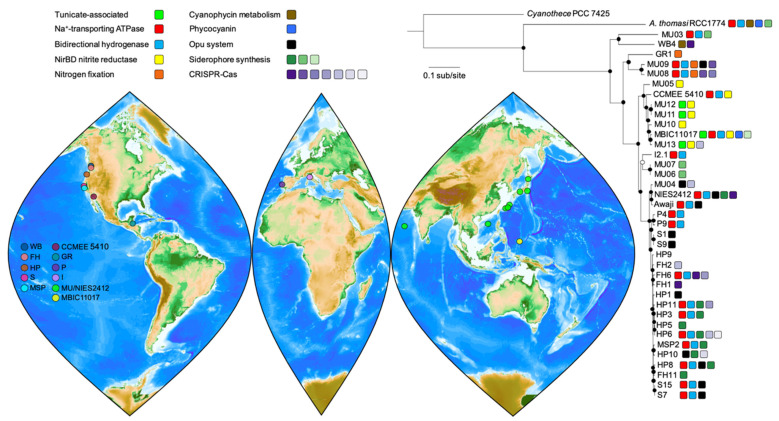
Sampling locations for *A. marina* strains and their evolutionary relationships. The genome-wide maximum likelihood phylogeny was reconstructed for a concatenation of 1369 protein sequences from single-copy orthologs according to the JTT+F+R5 model of sequence evolution and outgroup-rooted with *Cyanothece* sp. PCC 7425. Branch lengths are in units of the expected number of amino acid substitutions per site. Bootstrap support values of 100% (closed circles) and >90% (open circle) are shown for 1000 ultrafast bootstrap replicates. Selected traits discussed in the text are color coded as indicated. For the map of collection sites, note that the markers for the S. China Sea and Arabian Sea strains are approximate, as the exact sampling locations are unknown.

**Figure 2 microorganisms-10-00569-f002:**
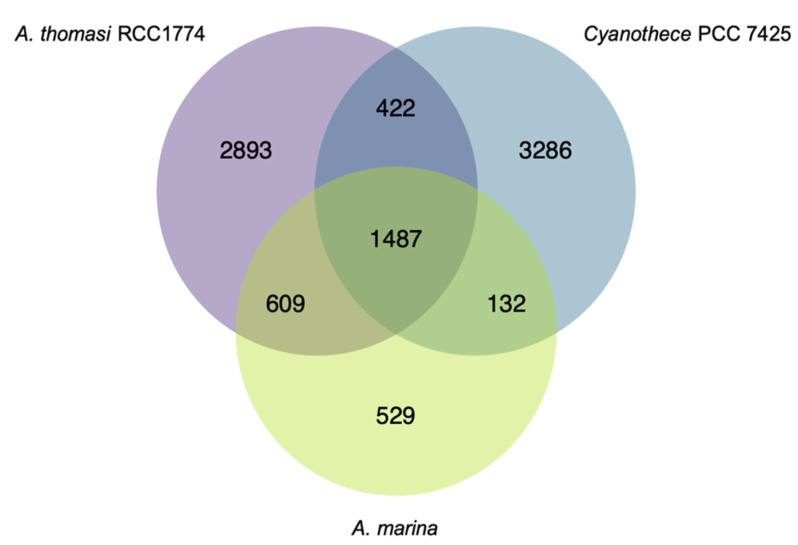
Venn diagram of shared and idiosyncratic protein-coding gene content among the *A. marina* core genome, the *Acaryochloris thomasi* RCC1774 genome, and *Cyanothece* PCC 7425 genomes. The diagram was generated with the R package VennDiagram.

**Figure 3 microorganisms-10-00569-f003:**
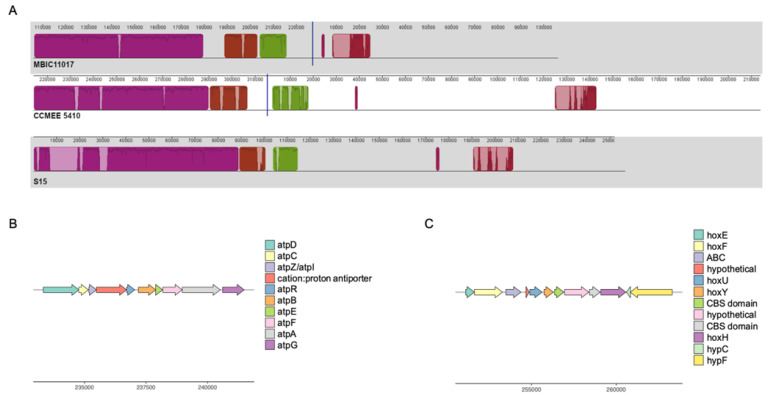
(**A**). Multiple alignment of plasmids from *A. marina* strains MBIC11017, CCMEE 5410, and S15 encoding sodium-transporting ATPase and bidirectional hydrogenase genes. Homologous blocks of aligned sequence share the same color, with missing DNA transparent. Traces within blocks indicate sequence similarity. (**B**). Gene map of the sodium-transporting ATPase region for *A. marina* CCMEE 5410. (**C**). Gene map of the bidirectional hydrogenase region for *A. marina* CCMEE 5410.

**Figure 4 microorganisms-10-00569-f004:**
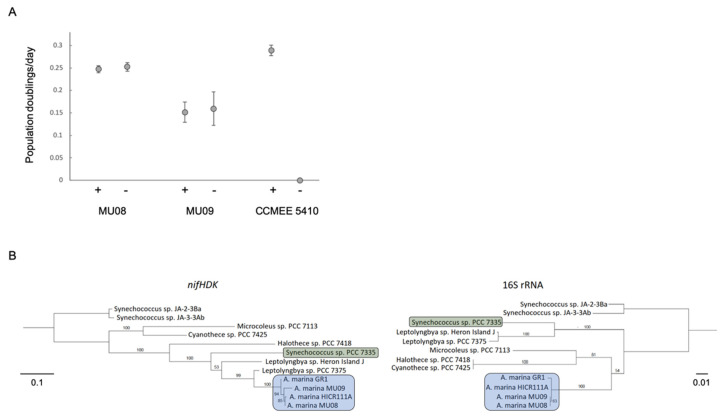
(**A**). Population growth rate in the presence (+) or absence (−) of combined nitrogen for *nif*-containing strains MU08 and MU09 and for *nif*-lacking strain CCMEE 5410. Error bars are standard errors. (**B**). Maximum likelihood phylogenies for *nifHDK* and 16S rRNA genes reconstructed with a GTR+I+G model. Bootstrap support values greater than 50% are shown for 1000 bootstrap replicates. Branch lengths are in units of expected number of nucleotide substitutions per site. The two topologies were significantly different by an SH test (*p* < 0.0001).

**Figure 5 microorganisms-10-00569-f005:**
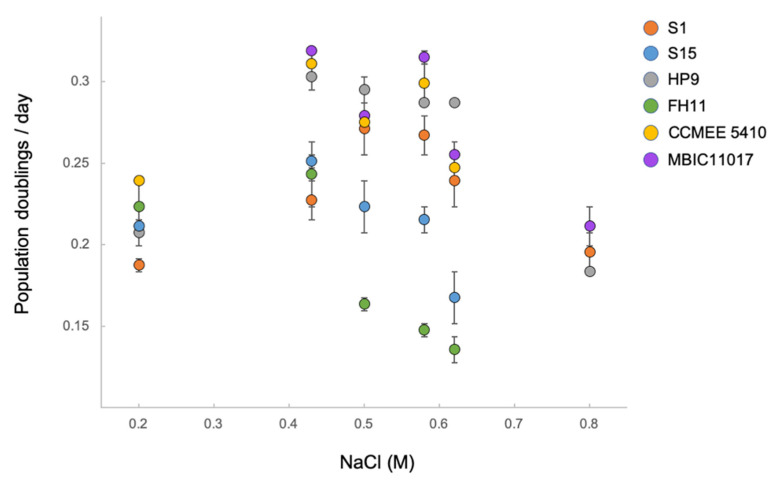
Population growth rates at different NaCl concentrations for upper intertidal *A. marina* strains S1 and HP9, subtidal strains S15 and MBIC11017, estuarine strain FH11, and saline lake strain CCMEE 5410. Error bars are standard errors. Note that 0.43 M is the NaCl concentration of standard marine ASN-III medium. Rates were estimated if growth was sustained for at least three generations. Pigments of strains for which growth was not observed at 0.2 M or 0.8 M NaCl did not bleach, whereas all strains bleached at 0 M and 1.6 M.

**Figure 6 microorganisms-10-00569-f006:**
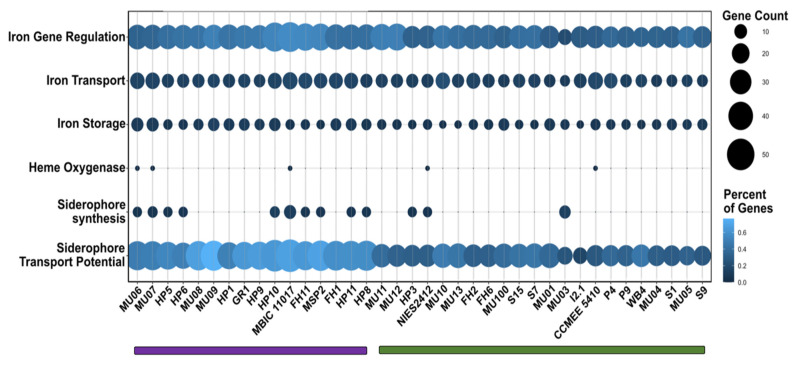
Distribution of genes involved in the regulation, acquisition, and storage of iron among *A. marina* strains based on both gene count and percent of genes in the genome. A dendrogram clustered strains into two groups based on high (purple) and low (green) total iron gene content. The approximately unbiased *p*-value of each cluster was 99%. See Appendix A for the hierarchical clustering dendrogram.

**Figure 7 microorganisms-10-00569-f007:**
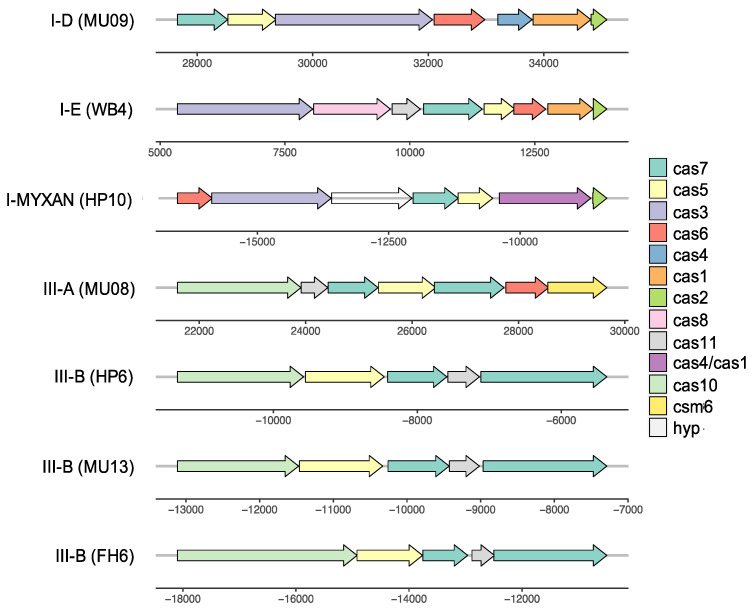
Representative gene maps for CRISPR-Cas systems detected in *A. marina* genomes (strain is indicated in parentheses).

**Table 1 microorganisms-10-00569-t001:** CRISPR-Cas systems detected in *A. marina* genomes.

Strains	Type	Repeat Consensus Sequence
MU08, MU09	I-D	GTTGCAAAAACGCTAAAACCCTCcAAGGGATTGAAAC
WB4, FH6, FH1, NIES2412	I-E	GTTGTCCCCACGCCTGTGGGGGTGGTCCG
HP10	I-MYXAN	AGCGGTGATTTAAGGTTTCCGGCCTGAAGCTTTGATGGACTT
MU08	III-A	GTTTCATCACTCATTCCCCGCAAGGGGACGGAAAC
HP6	III-B	GTTTCCAATAATTCCGATTGAAGTCAATCGGTAAAG
MU04, HP6, MU13, FH2	III-B	GTTTCCAATAATTCCGATTGAAGTCAATCGGTAAAG
HP11, FH6	III-B	GTTTTCATTTATTCGCCTTCCTACTGAATAGGAAG

## Data Availability

Genome data generated in this study can be found at NCBI BioProject PRJNA649288.

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
