# Peer review of "Genomic and Functional Variation of the Chlorophyll d-Producing Cyanobacterium Acaryochloris marina"

_microorganisms, 2022, doi:10.3390/microorganisms10030569_

Round 1
Reviewer 1 Report
Manuscript ‘Genomic and functional variation of the chlorophyll d-producing cyanobacterium Acaryochloris marina’ by Miller compared 37 strains of A. marina and identified 529 core genes that were not present in the outgroup cyanobacteria. A, marina spp. demonstrated enriched scavenging and storing iron metabolism according to genomic comparison. This manuscript provides genomic information for understanding functional variation of A. marina. The manuscript is written well. Authors could improve the genomic functional prediction by comparing the reported RNAseq data or other ‘omic’ analysis.
Author Response
Re: Authors could improve the genomic functional prediction by comparing the reported RNAseq data or other ‘omic’ analysis.
Response: Although this does not fall within the scope of this manuscript, we do plan to improve functional prediction using RNAseq data from another project in preparation.
Reviewer 2 Report
see attached
